# The effect of sleep on public good contributions and punishment: Experimental evidence

Jeremy Clark[1]*, David L. Dickinson[2,3,4]

1 Department of Economics and Finance, University of Canterbury, Christchurch, New Zealand,
2 Department of Economics and CERPA, Appalachian State University, Boone, North Carolina, United States of America, 3 IZA Institute of Labor Economics, Bonn, Germany, 4 Economic Science Institute, Chapman University, Orange, California, United States of America

* jeremy.clark@canterbury.ac.nz

**Data Availability Statement:** All relevant data are within the paper and its Supporting Information files.

**Funding:** This research was funded by the United States National Science Foundation BCS grant

## Abstract

We investigate the effect of a full week of sleep restriction (SR) vs. well-restedness (WR) on contributions in a common public good experiment, the voluntary contributions mechanism (VCM). We examine the effect of sleep manipulation on decisions regarding both contributions and punishment of non-contributors. Actigraphy devices are used to confirm that our random assignment to sleep condition generates significant differences in objective nightly sleep duration and sleepiness. We find that when punishment is unavailable public good contributions do not differ by SR/WR assignment. When punishment is available, we find evidence that SR subjects contribute more than WR subjects, respond more to the availability of punishment than do WR subjects, and that the availability of punishment significantly increases the contributions of SR but not WR subjects. Yet SR subjects do not punish others more or less than WR subjects. Our main findings are robust when considering compliance and sample selection. However, some findings are not robust to an alternative but less objective sleep control measure that is based partly on participants' self-identified optimal sleep levels.

## Introduction

According to the National Health Interview Survey, American adults' age adjusted average daily sleep fell from 7.40 hours in 1985 to 7.18 hours in 2004, holding steady at that level in 2012. Meanwhile, the proportion of adults sleeping less than 6 hours consistently increased, from 22.3% in 1985 to 29.2% in 2012 [1]. The US Center for Disease Control has recently labeled mild chronic sleep restriction a "public health problem." Some researchers suggest a bifurcation in sleep trends, with overall average sleep estimates remaining steady or rising, while the proportion getting less than 6 hours has increased [2]. This could be because sleep time is negatively correlated with both working hours and education, and over the past forty years individual labour hours have declined in the United States for less well-educated workers, but risen steadily for better-educated ones [3]. Internationally, the National Sleep Foundation's International Bedroom Poll and other studies have estimated insufficient sleep to be

number 1229067 and SES-1734137, awarded to
Prof. David Dickinson. This division of the granting
agency's website is https://www.nsf.gov/div/index.
jsp?div=BCS The funders had no role in study
design, data collection and analysis, decision to
publish, or preparation of the manuscript.

**Competing interests:** The authors have declared
that no competing interests exist.

habitual for 25%-50% of adults (https://sleepfoundation.org/sites/default/files/RPT495a.pdf, [4].)

Should society care about the tradeoffs people make between adequate sleep and other objectives? Counting direct effects on productivity and mortality risk, studies have estimated the economic cost of insufficient sleep to be anywhere between 1%-3% of annual GDP (see, for example, [4]). A recent cohort study found that insufficient sleep reduces participation in the labor force, particularly among low-skilled mothers [5]. As Costa-Font and Fleche note, this might suggest that children's sleep quality can affect a mother's sleep duration as a contributing factor to "poverty traps". The existing data and impact estimates also make it clear that, while acute total sleep deprivation matters, the majority of public health concerns are focused on the effects of more commonly experienced partial but chronic sleep deprivation. Experimentally, partial chronic sleep restriction (4–6 hours sleep per night) for 1 to 2 weeks has been shown to produce cognitive decrements similar to those caused by 1–2 nights of total sleep deprivation, though self-reported perceived sleepiness builds at a slower rate for the partially sleep restricted [6]. Those who are chronically partially sleep restricted may therefore be of greater public health concern given that awareness of one's deficits is likely linked to perceived sleepiness.

In this paper, we examine a potential novel spillover, or "external", effect of insufficient sleep on people's contributions to common or public goods. Joint contribution or cooperative dilemmas can involve real effort contribution decisions (such as housework, time spent volunteering or effort exerted in work teams), but also monetary contribution decisions regarding household and workplace public goods provision, or charitable donation decisions. Should employers (or economists) worry about negative externalities from encouraging longer working hours in our ever-connected age? Recent sleep research has explored causal links between sleep and some types of decision-making, but there is a gap in the literature regarding contributions to public goods even though such cooperative dilemmas abound in natural settings. We also aimed to study such public good dilemma environments both where contribution norm enforcement is possible, as well as when it is not possible. In the public goods literature this is often accomplished by examining a formal peer punishment mechanism. To our knowledge, this paper provides the first evidence of the impact of persistent mild sleep restriction vs. well-restedness on group cooperation and norm enforcement in a well-known social dilemma: the voluntary contributions mechanism (VCM) with peer punishment.

We report data from an experiment where university student subjects are randomly assigned to a full week of sleep-restriction (SR: 5–6 hrs/night attempted sleep) or required well-restedness (WR: 8–9 hrs/night attempted sleep) in their own homes. We use actigraphy ("sleep watch") devices common to sleep research to objectively measure sleep levels and assess compliance. At the end of the treatment week, subjects return to the lab and participate in a VCM experiment. We assess the validity of our sleep manipulation using objective and subjective measures in a completed sample of n = 126 subjects. For SR subjects, no limits were placed on compensating behaviors to combat sleepiness (other than additional sleep), which we think increases the external validity of our manipulation.

We measured sleep status in two ways: we primarily used binary random assignment (SR vs. WR) to treatment condition, but for robustness analysis we created a continuous sleep deprivation variable constructed from objective sleep levels *relative* to a participant's prior self-reported "optimal" sleep level, *Personal SD*. In addition, some of our estimations used an inverse probability weight correction for sample selection. Our most robust findings are that SR and WR subjects did not differ significantly in their contributions to the public good when costly punishment of others was unavailable. When punishment was available, there is evidence that SR's contributed more than WR's and had a greater contribution response to its

introduction. Yet, SR's and WR's did not differ in their likelihood or level of punishment towards others. Some results were sensitive to our use of *Personal SD* rather than binary SR assignment to control for sleep levels, or sensitive to conditioning our analysis on whether subjects met some standard of compliance with the prescribed sleep levels. These sensitivities, as well as some other suggestive results, are discussed in detail later. For example, one suggestive result was that 'anti-social' punishment (where subjects punish those who contribute more than they do), did not vary by binary SR/WR status, but increased in the continuous *Personal SD* measure.

The remainder of the paper works as follows. In Section 2 we review the literature relevant to VCM experiments with punishment, and the effects of sleep deprivation on social interactions. Section 3 provides our experimental design, while Section 4 provides our results. Section 5 concludes.

## Literature review

Public good provision and norm enforcement have received significant attention in the experimental economics literature. The VCM has served as a common template for the examination of behavior in cooperative dilemmas. Here, individual incentives are at odds with group incentives, and the common finding is that groups initially contribute 40%-50% of the socially optimal level to the group good, with contributions declining under finite repetition [7]. Subsequent investigations have identified factors that influence cooperative or free-riding behavior, such as Isaac *et al.* [8], Andreoni [9], Isaac and Walker [10], and survey papers by Ledyard [7] and Chaudhuri [11]. One prominent factor has been costly peer punishment as a device for norm enforcement. Ostrom *et al.* [12] first examined peer punishment in common pool resource dilemmas, while Fehr and Gächter [13] did so using the VCM. Fehr and Gächter found that punishment was used primarily against those contributing less than the group average, and that its introduction significantly increased cooperation. Subsequent studies confirmed the effectiveness of punishment in raising cooperation, particularly if peers punish *low* rather than *high* contributors ("pro-social" rather than "anti-social" punishment) [14, 15]. For example, Gächter *et al.* [16] tested whether punishment reduced net earnings given the offsetting costs incurred by punishers and the punished. They found that punishment was welfare enhancing in a repeated game with sufficiently long horizon. Peer punishment has been investigated also by Masclet *et al.* [17], Noussair and Tucker [18], Bochet *et al.* [19], Anderson and Putterman [20], Sefton *et al.* [21], Egas and Riedl [22], Nikiforakis [23], Nikiforakis and Normann [24], Engelmann and Nikiforakis [25], and Dickinson and Masclet [26].

In contrast to the extensive VCM literature, there are few studies of the effect of sleep on economic decisions. We know of none that have examined the effect of sleep manipulation on voluntary contributions to public goods. However there have been studies of the effects of sleep manipulation on decisions in other social tasks. Anderson and Dickinson [27] found a single night of *total* sleep deprivation (TSD) reduced subjects' revealed trust in a standard trust game, and increased the minimum offer that responders would accept in ultimatum bargaining. Both effects are consistent with sleep deprivation increasing one's aversion to being exploited by others, which may have implications for VCM behavior. Ferrara et al. [28] examined the effect of one night TSD on dictator decisions and found it reduces giving, though only among females. More relevant is a study by Dickinson and McElroy [29] on sleep restriction and decisions in 2-player games. Dickinson and McElroy examined a larger sample of subjects compared to typical TSD research, and prescribed partial at-home sleep restriction over a full week rather than single night TSD. Arguably, chronic partial restriction is more relevant to examining sleep effects outside the lab, where people regularly get insufficient sleep, rather

than one night without sleep. Dickinson and McElroy found SR reduced subjects' prosocial behaviors in general (including trust, trustworthiness, and altruistic giving), which again may have implications for VCM outcomes. While these authors did not find sleep affected ultimatum bargaining outcomes, this result is of less interest given the confounding reasons why proposers there may make generous offers (i.e., generosity versus fear of rejection of offer). A somewhat related literature has examined whether cooperation is more likely with an intuitive or a deliberative approach to decision making. Initial studies suggested cooperation was more intuitive [30, 31]. However, these findings were not without challenge (e.g., [32]), and replication efforts have shown that the time-pressure methodology used to induce "intuitive" decisions may have caused sample selection that biased results [33].

Unfortunately, these limited results provide inconclusive predictions for the direction of effect of sleep restriction on VCM contributions or punishment. If contributions are viewed as prosocial, then Dickinson and McElroy's findings suggest SR will decrease contributions. Yet any increased aversion to punishment may predict that SR will increase contributions. With regard to punishment, we are not aware of any studies regarding SR effects. Anderson and Dickinson's [27] argument that sleep deprivation increases people's aversion to exploitation (or suffering loss in social exchange) might suggest it will make them punish more. However costly punishment may be considered an expression of anger and revenge at free-riders, or an altruistic and rational form of pro-social norm enforcement [13]. If it is the former, SR has been shown to increase conflict in couples [34], worsen mood, and increase irritability [35–37], all predicting that SR will increase one's proclivity to punish others. Dickinson and Masclet [26] identify such emotion-driven punishment as detrimental to overall VCM outcomes. Yet if punishment is the latter, neuroscience has found that deliberative thinking is important for prosocial decisions (see [38–40]) such as altruistic punishment. Since sleep deprivation is known to disproportionately affect people's deliberative (prefrontal) brain activation [15, 41, 42] SR may decrease their proclivity to punish others out of deliberative altruism. Consistent with this, Dickinson and McElroy [29] note some studies have found increased prefrontal activation in sleep deprived subjects, but this was not found to enhance decision-making. Sleep deprivation has been found to increase "ventromedial prefrontal cortex" activation, which may suggest an enhanced focus on potential for monetary gain and potential for optimism bias (see [43]).

Our research is therefore exploratory in that we do not have unambiguous *a priori* hypotheses regarding the effect of reduced SR vs. WR on costly contributions to the public good or on norm enforcement.

## Experimental design

Prior to the experiment, the project was reviewed and approved by the Institutional Review Board at Appalachian State University (approval number 16–0067) under the title "Sleep Restriction and Decision Making." Informed consent was obtained in writing from each participant prior to the beginning of the study. All participants were 18 years of age or older and so able to give their own consent to participate. We initially administered an online screening survey to generate a database of several hundred potential subjects for recruitment to this study. The survey contained a short validated "morningness-eveningness" questionnaire to assess diurnal preference [44], commonly used general practitioner screening questions for depression and anxiety disorders, and some self-reported sleep measures. We excluded from recruitment subjects scoring at risk for a major depressive or anxiety disorder, or who reported a sleep disorder or insomnia. We also excluded subjects with strong morning- or evening-type preference so as not to introduce the confounding factor of circadian (mis)timing of our

subsequent decision tasks. We also controlled for circadian timing effects by running all decision-making sessions between 10am-4pm, and from Tuesday-Thursday to minimize weekend effects.

Once ineligible subjects were removed, remaining subjects were randomly assigned, *ex ante*, to the WR (8–9 hr/night) or SR (5–6 hr/night) treatment condition. Note that our protocol assigned participants either to; (a) one week of sleep levels similar to typical recommendations for good sleep hygiene, or to (b) a restricted sleep level deemed concerning yet commonly experienced by millions of adults. These assigned sleep levels increase the applicability of our findings to real world decision makers. If adhered to, these levels would result in WR-sleep above and SR-sleep below average levels of about 6.5 hrs/night observed among comparable undergraduate experiment participants by Dickinson *et al.* [45].

After random treatment assignment were made, recruitment emails were then sent to subjects inviting them to participate in a one-week experiment that would involve a prescribed nightly sleep level for 7 nights and culminate in a 1.5 hour decision session. The recruitment email specified in detail the sleep condition assigned to the subject. Subjects were not allowed to opt out of one sleep treatment to select the other; they either participated in their assigned sleep condition, or did not participate.

Subjects were also informed they would be required to wear a wrist-watch sized actigraphy device to objectively yet passively measure their sleep levels, and keep a sleep diary. The actigraphy device is a wrist-worn accelerometer intended to be worn all day with few exceptions. It is common to sleep research and has several advantages over lower cost commercial devices. Its validity has been established in the sleep research literature and is a well-accepted way to generate objective, valid data on sleep duration in non-disordered individuals [46] as well as some young adult clinical populations (e.g., insomniacs, [47]). Additionally, the use of actigraphy technology for passive sleep data acquisition is part of the accepted methodology outlined in the joint consensus statement of the American Academy of Sleep Medicine and the Sleep Research Society on recommended adult sleep levels [48].

We used research-grade devices (Actiwatch Spectrum Plus, from Philips), and followed standard scoring protocols—see those established by the Society of Behavioral Sleep Medicine that are used by clinicians as well as researchers [49]. Importantly, where an experimental sleep manipulation is introduced, recent evidence has shown the clear benefit of objective actigraphy data due to bias in total sleep times people self-report in sleep diaries [37].

For our one-week at-home sleep protocol, subjects were required to make two lab visits. At Session 1 they answered questions on a 6-item cognitive reflection task [50], a short-version of the Big Five personality measure [51], and were then issued the actigraphy device, sleep diary and instructions for the week. Subjects were then free to ask questions, which were answered without revealing any individual's assigned treatment. Cohorts of typically 10–15 subjects were recruited for a given week, and were a mix of SR and WR.

Upon leaving the first session, subject contact with experimenters was limited to daily text or emails to indicate bed/wake times. Subjects also recorded this information in their sleep diaries, but the emails allowed the experimenter to monitor their attempted sleep levels. Notwithstanding the self-reported emails and sleep-diaries, we prioritized the actigraphy-recorded sleep data for each subject. Subjects were also emailed every 1–2 days to remind them of the prescribed sleep levels, caution them regarding the risk of certain activities when sleepy (sent to both SR and WR subjects), and remind them of the upcoming session. Session 2 occurred one week after session 1 at the same time of day, and included a short survey and self-report on sleepiness, two decision experiments, the removal of the actigraphy devices, and cash payments based on decisions made. In addition, subjects also received a fixed $25 for adhering to the conditions of the sleep protocol and returning the actigraphy device and sleep diaries at

session 2. Subjects were informed they would receive the fixed payment several days later by Amazon.com gift code or check (their choice), after sleep data were downloaded and the experimenters could verify good faith compliance efforts. Our threshold for "compliance" to pay subjects the $25 was not as stringent as our compliance standard for subsequent data analysis. We wished to err on the side of paying subjects the $25 in most instances and gave partial payment to the few subjects who withdrew partway through the sleep treatment week.

For the public good task, we first followed established protocol in setting up a typical VCM cooperative dilemma that pits self-interest against group interest, because this serves as a standard building block group task for understanding behavior in more complicated real world cooperative decision environments. Subjects were randomly assigned to groups of 3, each endowed with 15 tokens, and then asked to decide how many tokens to 'keep' or 'invest' towards joint production. Each token kept was worth $.08 in experimental currency to that subject alone, while each token invested was worth $.05 both to that subject and to each of the other 2 group members. Similarly, each token invested by other group members generated $.05 for that subject. In what follows, we refer to investments as "contributions", as is common in the literature. These parameters create a marginal per capita return (MPCR) on investment of .625, which provides the standard dominant strategy to free ride off others' contributions even though full contributions maximize total earnings. The exchange rate was announced verbally at each session, and was Experimental Dollars $1.00 = US$0.40 for all but the first of nine cohorts, where it was US$0.50.

With our interest in testing for the effects of sleep manipulation in cooperative dilemmas without and with norm enforcement opportunities, we included a treatment that allowed for peer punishment (again, following standard protocol in the VCM literature were punishment has been studied). Subjects played in a *partners* (fixed group) design for a 10 round treatment that included a punishment option (Punishment), and a 10 round treatment without a punishment option (No Punishment). The partners matching design choice allowed for reputational and person-specific learning effects with groups. Cohorts 1, 3, 7, 8 and 9 experienced the "No Punishment" treatment first, and cohorts 2, 4, 5 and 6 experienced the "Punishment" treatment first. Subjects were informed of each individual's contribution to the group account after each round. While subjects were aware of the fixed matching protocol, interactions took place anonymously over a computer network using the Veconlab VCM software platform. The details of the game can be found at http://veconlab.econ.virginia.edu/pg/pg.php. Thus subjects did not know the sleep status of others in their group, and groups contained both SR and WR subjects. We chose this design to parallel real world applications where individuals are not aware of the sleep status of others with whom they face collective contribution dilemmas, and where such dilemmas do not emphasize the sleep status of group members.

When peer punishment was available we used a linear cost structure as available in the Veconlab software implementation of this game. After subjects were informed of the total group contributions for a round, each group member could elect to assign "punishment points" to each of the other two group members (up to a total of 10 assigned points assigned by a group member toward others). Each point sent cost the sender $0.10, so that one could spend up to $1 punishing others each round. Punishment sent was funded from stage 1 earnings. Subjects were always able to send up to 10 punishment points; in the rare cases where a subject earned less than $1 in stage one, the senders' net earnings for that round could be negative. This happened in only 2 instances out of over 1000 decisions in Punishment rounds.

Each point received cost the recipient 10% of his/her earnings that round. While in principle, someone could lose as much as 200% of first stage earnings if he or she received 20 punishment points, subtractions were capped at 100% of earnings, with senders' costs reduced proportionately. In practice, this capping occurred in only 5 of 1260 cases. Subjects were

informed of the ID number of the person sending punishment points. The potential for 'round bankruptcy' was offset by giving all subjects $5 prior to Round 1.

## Results

All sessions were run at Appalachian State University from March-November of 2016. Average VCM earnings (in US$) were $14.57 ± $2.31 (min = $8.94, max = $20.80) paid in cash at the end of the session. As noted above, subjects were also paid $25.00 for sleep protocol compliance, which was sent a few days after the session. Each cohort's Session 2 included another decision task for which subjects earned cash, although no payments were given until the end of the session (and order of tasks was varied across sessions).

### Attrition and compliance

We initially recruited N = 167 subjects, but not all completed the one-week sleep protocol. 16 failed to show up for the first session (6 SR, 10 WR), and 18 withdrew during the week (17 SR, 1 WR). The fact that almost all who dropped out during the protocol were SR was likely due to subjects having found compliance with SR more difficult than they anticipated. Interestingly, of those subjects who completed the one-week protocol, most who were found non-compliant *ex post* were WR subjects. That is, rather than withdraw from the study as non-compliant SR subjects did, non-compliant WR subjects were more likely to finish the protocol but not achieve sufficient rest. We therefore had 133 subjects who completed the one-week protocol and VCM experiment. Of these, 7 experienced sleep watch malfunctions or failed to wear the device. Thus, our complete sample of sleep and behavioral data consisted of 126 treatment subjects (61 SR, 65 WR; n = 81 female). We discuss how we deal with the risk of selective attrition shortly.

   Aside from attrition, an additional concern in a sleep manipulation study is compliance with the randomly assigned sleep level. Fig 1 shows the distributions (kernel density estimates) of nightly sleep for our completed 126 treatment subjects as objectively measured by actigraphy, and clearly reveals that not all subjects fully complied with their prescribed sleep levels. This presents us with a dilemma when constructing our main binary SR/WR classifications. We are interested in the effects of *actual* sleep manipulation on joint production, and thus in the decisions of subjects who "complied" with sleep protocols. But we faced a judgement call over how strict our definition of "compliant" vs. "noncompliant" should be. Following previous studies, we chose data-driven thresholds to classify SR subjects as compliant if their actigraphy data averaged 375 minutes (6.25 hours) or less of nightly sleep, and WR subjects as compliant if they averaged 405 minutes (6.75 hours) or more. For comparison, using a within-subject protocol, Dickinson *et al.* [37] also used a compliance standard that was subjective but data driven and based on the desire to minimize the likelihood that a week duration sleep restricted subject was statistically indistinguishable from a control subject. Reassuringly, the excluded region (6.25 to 6.75 hrs/night sleep) is close to the average nightly sleep levels experienced by adults according to recent surveys (Dickinson *et al.* [45], and the National Sleep Foundation 2013 *International Bedroom Poll* at https://sleepfoundation.org/sites/default/files/RPT495a.pdf). Recent Gallup poll results have found that average sleep levels of younger adults were lower than adults in general (see http://www.gallup.com/poll/166553/less-recommended-amount-sleep.aspx [accessed March 31, 2017]). Thus the average sleep levels of adults the age of our college student sample are likely similar to our noncompliance range of sleep. Excluding those with near average nightly sleep levels can be thought of as a conservative way to remove subjects who are indeterminate to classify. We then condition our main analysis on compliance. As can be seen in Fig 1 (and in Table 1), this filter removes 17 non-

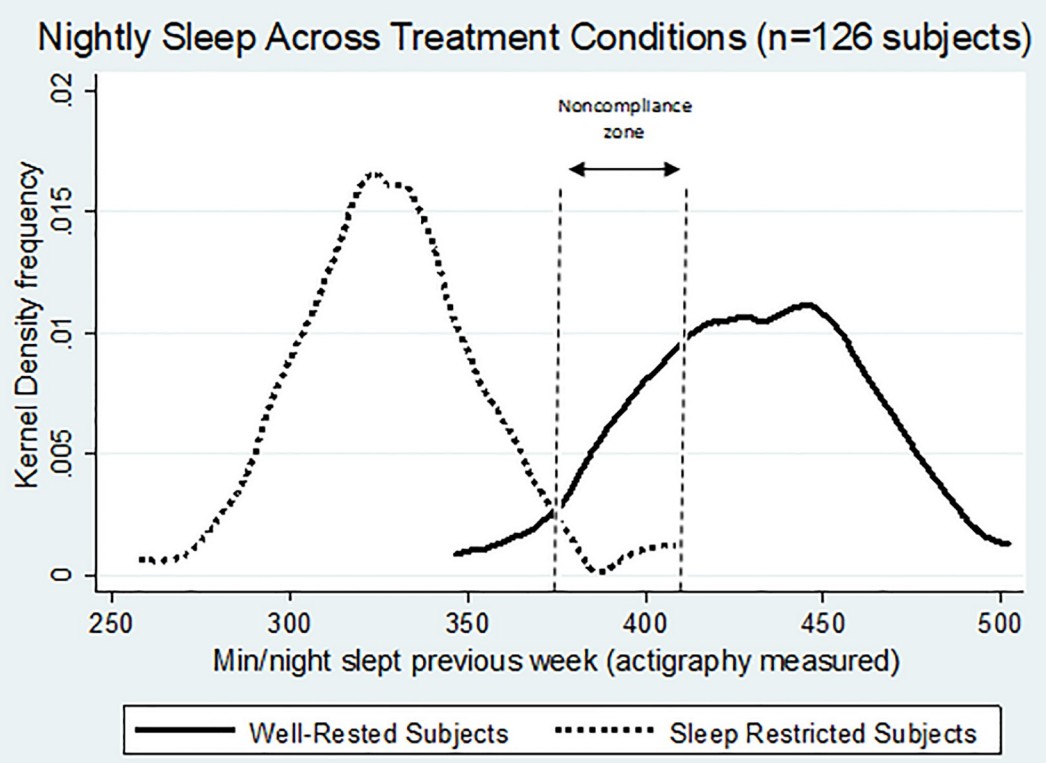

**Fig 1. Sleep levels by treatment assignment.**

compliant subjects, but predominantly from the WR assignment (n = 15), which leaves us with n = 109 subjects scored as compliant (59 SR, 50 WR). This is a similar compliance rate as that obtained in a more extensive at-home sleep protocol [37]. Note that in total, we lose somewhat similar numbers of assigned SR's and WR's to the combined issues of withdrawal and noncompliance (n = 19 SR vs. n = 16 WR). However as mentioned, SR subjects tended to voluntarily withdraw during the protocol week, whereas WR subjects finished the week but with a lower compliance rate. SR subjects may have underestimated how difficult it would be to complete the restricted sleep protocol, while WR subjects may have underestimated the degree to which they needed to modify their schedules to actually sleep more than usual (or overestimated how efficient their nightly sleep was). Descriptive statistics by treatment and compliance are presented in Table 1. Due to the high fixed costs of recruiting subjects for the sleep protocol, we ensured full use of all treated subjects by recruiting a small number of "backup" subjects with no prior sleep manipulation for each decision session. These subjects were recruited to ensure the total number of subjects in the session was divisible by 3. Of the 49 total VCM groups containing 147 subjects, 16 were backups. We did not analyze the decisions of the backup subjects, but we did analyze the behavior of SR and WR subjects in groups with them. We believe this to be valid because group members were not aware of the sleep status of others in their groups. Balance tests showed no significant differences in age, gender, or cognitive reflection scores across the SR-WR treatment assignment (see Fig 1 in S1 Appendix).

The reduction in our sample size from N = 167 to 126 subjects with full data, and 109 compliant subjects, raises concerns regarding selective attrition and lack of statistical power to discern small magnitude treatment effects. We address attrition as follows. First, we conducted all our analysis both on our compliant sample (N = 109), but also on our "intent-to-treat" sample

**Table 1. Descriptive statistics.**

| A: Compliant Subjects Only | | | | | | |
|---|---|---|---|---|---|---|
| Variable | Assigned Treatment | N | Mean | St Dev | Min | Max |
| Age | Sleep Restricted | 59 | 19.88 | 1.80 | 18 | 28 |
| | Well Rested | 50 | 20.88 | 4.58 | 18 | 43 |
| | Total | 109 | 20.34 | 3.39 | 18 | 43 |
| Female | Sleep Restricted | 59 | 0.66 | 0.48 | 0 | 1 |
| | Well Rested | 50 | 0.62 | 0.49 | 0 | 1 |
| | Total | 109 | 0.64 | 0.48 | 0 | 1 |
| Cognitive Test[1] | Sleep Restricted | 59 | 37.01 | 30.80 | 0 | 100 |
| | Well Rested | 50 | 37.33 | 28.68 | 0 | 100 |
| | Total | 109 | 37.16 | 29.71 | 0 | 100 |
| Personalized | Sleep Restricted | 59 | 160.45 | 61.95 | -34.5 | 333.07 |
| Sleep Deprivation[2] | Well Rested | 50 | 47.08 | 53.64 | -82.07 | 164.50 |
| (in minutes/night) | Total | 109 | 108.44 | 81.16 | -82.07 | 333.07 |
| B. Compliant and Noncompliant Combined | | | | | | |
| Variable | Assigned Treatment | N | Mean | St Dev | Min | Max |
| Age | Sleep Restricted | 61 | 19.90 | 1.80 | 18 | 28 |
| | Well Rested | 65 | 20.72 | 4.10 | 18 | 43 |
| | Total | 126 | 20.33 | 3.21 | 18 | 43 |
| Female | Sleep Restricted | 61 | 0.67 | 0.47 | 0 | 1 |
| | Well Rested | 65 | 0.62 | 0.49 | 0 | 1 |
| | Total | 126 | 0.64 | 0.48 | 0 | 1 |
| Cognitive Test[1] | Sleep Restricted | 61 | 37.43 | 30.98 | 0 | 100 |
| | Well Rested | 65 | 38.21 | 28.98 | 0 | 100 |
| | Total | 126 | 37.83 | 29.84 | 0 | 100 |
| Personalized | Sleep Restricted | 61 | 155.61 | 67.36 | -48.29 | 333.07 |
| Sleep Deprivation[2] | Well Rested | 65 | 58.23 | 58.92 | -82.07 | 193.21 |
| | Total | 126 | 105.37 | 79.64 | -82.07 | 333.07 |

[1] A cognitive reflection task scored over 6 questions where heuristic short cuts might suggest an incorrect answer (see Primi et al. 2015). Can range from 0 to 100.

[2] Subtracts Previous Week's Sleep from the subject's self-reported level of optimal sleep (both in minutes per night).

that includes both the compliant and non-compliant (N = 126). Our latter results are given in S1 Appendix Tables. Second, we employed the method of inverse probability weighting (*IPW*) to correct for selection in attrition. Under *IPW*, we first ran a probit selection regression based on observable characteristics we had on the entire N = 167 sample from the online preliminary screening survey recorded for the entire pool of potential recruits (e.g., SR/WR assignment, gender, minority status, age, self-reported optimal sleep level, and other sleep related variables —see Table 2 in S1 Appendix). We used this regression to predict the probability of individual subjects persisting through to the completed VCM task (N = 126). By assigning the inverse of this probability as a weight to each subject in the intent-to-treat (N = 126) or compliant (N = 109) samples in weighted regressions, we increased the weight given to the observations of those subjects who were in our final sample but whose observable characteristics predicted a higher probability of being lost to attrition. In the estimations that follow, we present estimation results from the compliant (N = 109) sample in the main text with corresponding analysis using the full intent-to-treat (N = 126) sample in the S1 Appendix. In all cases, we tried specifications that use the *IPW* correction for comparison with the uncorrected specification results. Reassuringly, our estimation results are largely unchanged when using the *IPW* correction.

Turning to the issue of statistical power, it is possible that tests using our complete (N = 126) or compliant (N = 109) samples may lack the power to detect smaller magnitude treatment effects. Yet the resource-intensive nature of recruiting and monitoring subjects (i.e., device availability) precluded running further sessions. To investigate this, we examined the predicted power of our design using an *ex ante* power analysis of the detectable effect size for our compliant (N = 109; 59 SR, 50 WR) or complete sample (N = 126: 61SR, 65 WR) with power $(1-\beta)$ = .80 and $\alpha$ = .05 in two tailed tests using G*Power (version 3.1.9.4). Using Mann Whitney tests of each individual's average contributions over all rounds of a given punishment regime, we find a standarized effect size (*d*) as small as .556 would be detectable for the compliant sample, or .515 for the complete sample. With our sample sizes we should thus be able to detect approximately "medium" sized effects (Cohen's *d* = .5), but not "small" sized main effects (e.g. *d* = .2) using Mann Whitney tests. Slightly more optimistically, our parametric main Tobit regression models using the 109 subjects' 20 individual round decisions (pooled cross section with standard errors clustered to group) will presumably be able to detect a somewhat smaller treatment effect size. However, we could not readily conduct power analysis for such regression models to identify exact thresholds.

Attrition and power aside, we next assessed the success of our protocol at manipulating sleep using objective and subjective measures. Our objective measure was actigraphy-derived sleep levels during the treatment week, *Previous Week Sleep*, in average number of minutes of sleep per night. Another quasi-objective measure, *Personal Sleep Deprivation* (or, *Personal SD*) was constructed by subtracting *Previous Week Sleep* from the subject's self-assessed sleep need. The self-perceived sleep need measure was elicited earlier in the preliminary online screening survey, and so it poses no risk of endogeneity due to the subject's subsequent random assignment.

*Personal SD* has the advantage of recognizing that some individuals need more sleep than others, but assumes people know their self-defined "optimal" sleep levels. Subjects also reported their sleepiness on a 1–9 scale (9 = most sleepy), *Karolinska Sleepiness*, and reported the extent to which the protocol made them sleep less or more than typically (ranging between -4 to +4, where 0 was "no effect"), *Treatment Impact*. These two subjective measures were elicited at the end of the sleep treatment week, before the decision task. Table 2 reports manipulation validity checks for each of these measures, using both our compliant and intent-to-treat samples. Reassuringly, Mann-Whitney tests showed a significant difference between the SR and WR groups using all these measures ($p$ < .01 in all instances). We also note there was no

**Table 2. Compliance with sleep protocol (objective and subjective measures) means (st dev) in minutes.**

| Variable | SR assigned (n = 61) | WR assigned (n = 65) | Mann-Whitney test (WR-SR) Z-statistic |
|---|---|---|---|
| *Previous week sleep* (min/night) | 328.32 (.32) | 430.54 (32.14) | 9.43 ($p$ < .01) |
| *Personal SD* (min/night) | 155.61 (67.36) | 58.23 (58.92) | -7.16 ($p$ < .01) |
| *Karolinska sleepiness* [1–9] scale | 6.03 (1.84) | 3.58 (1.40) | -6.71 ($p$ < .01) |
| *Treatment Impact* [-4, +4] scale | -2.87 (1.04) | 1.44 (1.51) | 9.59 ($p$ < .01) |
| Variable | SR compliant (n = 59) | WR compliant (n = 50) | Mann-Whitney test (WR-SR) Z-statistic |
| *Previous week sleep* (min/night) | 325.65 (22.71) | 443.12 (23.77) | 8.97 ($p$ < .01) |
| *Personal SD* (min/night) | 160.45 (61.95) | 47.08 (53.64) | -7.52 ($p$ < .01) |
| *Karolinska sleepiness* [1, 9] scale | 6.12 (1.78) | 3.75 (1.43) | -6.08 ($p$ < .01) |
| *Treatment Impact* [-4, +4] scale | -2.92 (1.00) | 1.54 (1.47) | 8.88 ($p$ < .01) |

**Notes:** Two non-compliant subjects assigned to SR and one assigned to WR did not provide actigraphy data but provided other sleepiness measures. Including these subjects would lead to n = 63 SR and n = 66 WR subjects. Table 1 measures and tests for *Karolinska sleepiness* and *Treatment impact* are not appreciably affected by the exclusion or inclusion of these subjects.

significant difference in the self-reported optimal sleep levels between the 167 participants sub-sequently assigned SR or WR conditions (Mann-Whitney, $p = .61$), nor between the SR's and WR's who completed the sleep protocols and VCM (N = 126, $p = .53$), nor between those who completed and complied with the protocols (N = 109, $p = .61$). Given that the probit selection equation in Table 2 in S1 Appendix indicates that subjects with higher "optimal sleep" need were less likely to complete the study, this lack of difference between SR and WR subjects at all three points suggests that the higher attrition among those with higher perceived sleep need was not systematically biased towards one sleep assignment or the other.

As foreshadowed, the variable *Personal SD* has use beyond manipulation checks because it represents an alternative continuous measure of inadequate sleep that is both personalized to perceived sleep need, yet still partially anchored to objective sleep measurement. In what fol-lows, we shall report results using both the binary SR/WR assignment control as well as the continuous *Personal SD* measure.

## Contribution decisions

Of the nine cohorts of subjects facing the VCM after the one week protocol, we implemented a slight change after cohort 1, and a problem emerged with cohort 3. Following higher than anticipated earnings for cohort 1, we lowered the exchange rate from experimental currency to US dollars from $0.50 to $0.40, keeping unchanged the relative incentives between keeping and investing tokens. Of greater concern, a programming error for cohort 3 led to a difference being introduced between the return an invested token yielded to the investor vs. to other group members. Specifically, in cohort 3 the return to an individual from keeping a token was $.10 (instead of $.08), and the return from investing it was $.08 for oneself but $.05 for each of the other 2 group members. While the (money maximizing) dominant strategy remained com-plete free-riding and efficiency full contribution, the exact MPCR differed, as did the complex-ity of the incentives. For preliminary non-parametric tests, we thus present results with and without cohort 1 (since the exchange rate change did not affect relative incentives), and we test whether or not cohort 3 can be pooled with the other cohorts. These pooling tests for cohort 3 found that subjects differed in punishment sent when available ($p$-value = .005 in two-tailed Mann-Whitney tests, using individual ten-round averages), and we thus excluded it. For cohort 1 subjects, though we do not believe the change in exchange rate was at fault, we note that WR subjects contributed unusually little to the public good, such that contributions dif-fered significantly when punishment was unavailable ($p$-value = .048). Fortunately, while our non-parametric analysis will thus exclude cohort 3 and show results with and without cohort 1, we will be able to retain both cohorts in our regression analysis by including separate cohort 1 and 3 dummy variables, as well as sleep treatment interaction terms with each of these cohorts.

Fig 2 illustrates average individual contributions across rounds (using compliant individu-als from all nine cohorts) by sleep and punishment treatment. From panel (a), SR and WR sub-jects appeared to contribute similarly without punishment, whereas SR's seemed to contribute more than WR's with punishment available. We note cohort 1 had an especially pronounced disparity in SR/WR contributions because WR's contributed less. From panels (b) and (c), punishment availability seems to modestly raise contributions overall, due almost entirely to its effect on SR's. To test for effects more formally, we next move to nonparametric and regres-sion analysis.

Table 3 reports the 10-round average contributions of (compliant) individuals by sleep and punishment treatment, and Mann-Whitney tests for sleep effects (Table 1 in S1 Appendix pro-vides analogous results including non-compliant subjects). As foreshadowed in Fig 2(A),

### a. Sleep Restricted (SR) vs. Well Rested (WR)

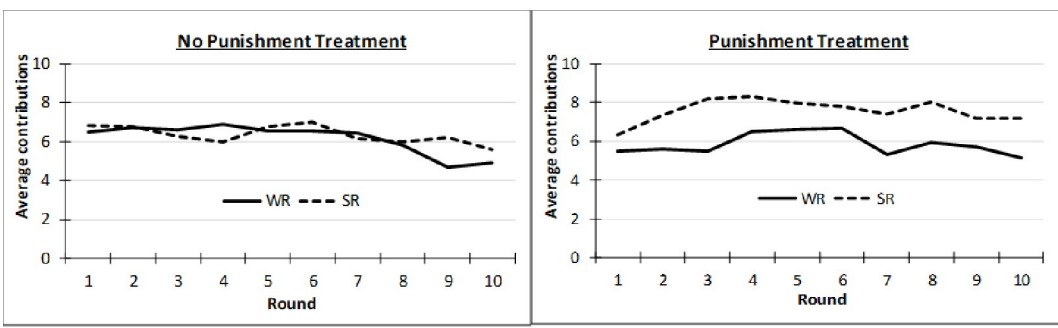

### b. Punishment vs. No Punishment (Pooled)

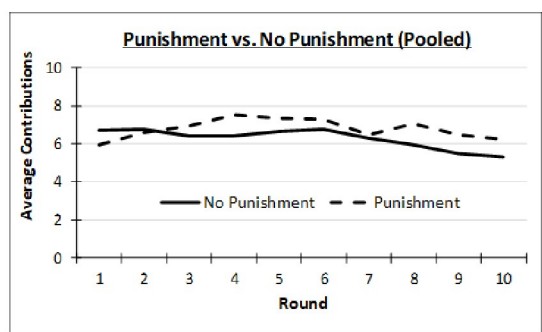

### c. Punishment vs. No Punishment by Sleep Condition

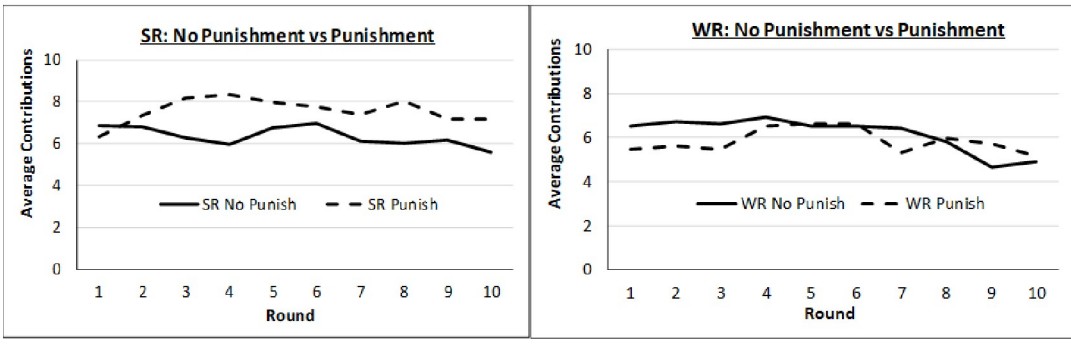

**Fig 2. Average contribution levels (Compliant subjects only).**

average contributions from SR and WR subjects were similar without punishment. Averaged over ten rounds, SR contributions were 6.58 tokens (6.53 tokens with cohort 1), while WR contributions were 6.44 tokens (or only 5.83 tokens with cohort 1). Not surprisingly, without punishment, a lack of difference in contributions cannot be rejected (two-tailed $p$-value = .797 without cohort 1, or .554 with it). When punishment is available, mean SR contributions climbed to 7.49 tokens (7.58 tokens with cohort 1), while WR contributions fell slightly to 6.27 tokens (5.74 tokens with cohort 1). Yet despite these differences in mean contribution by sleep condition, the standard deviations were high enough that the difference was not significant (two tailed $p$ = .303), though marginally significant with cohort 1 included ($p$ = .054).

**Table 3. Non parametric tests for treatment effects—compliant subjects only[a].**

*Contributions* (Averaged Over 10 Rounds)

| | Excluding Cohort 1 | | | Including Cohort 1 | | |
|---|---|---|---|---|---|---|
| Punishment Not Available | | | | | | |
| | Sleep Restricted | Well Rested | Mann Whitney SR vs WR p value | Sleep Restricted | Well Rested | Mann Whitney SR vs WR p value |
| Mean | 6.58 | 6.44 | | 6.53 | 5.83 | |
| Stand Dev | 4.57 | 3.64 | 0.797 | 4.42 | 3.75 | 0.554 |
| N | 47 | 36 | | 53 | 42 | |
| Punishment Available | | | | | | |
| Mean | 7.49 | 6.27 | | 7.58 | 5.74 | |
| Stand Dev | 4.02 | 3.87 | 0.303 | 4.06 | 3.96 | 0.054* |
| N | 47 | 36 | | 53 | 42 | |
| *Punishment Sent* (Averaged Over 10 Rounds) | | | | | | |
| | Sleep Restricted | Well Rested | MannWhitney SR vs WR p value | Sleep Restricted | Well Rested | Mann Whitney SR vs. WR p value |
| Mean | 0.82 | 0.54 | | 0.73 | 0.55 | |
| Stand Dev | 1.51 | 0.73 | 0.466 | 1.44 | 0.69 | 0.960 |
| N | 47 | 36 | | 53 | 42 | |

[a] In all cases excluding cohort 3, for which there were programming errors

We recognize that, while Mann Whitney tests of medians are illustrative, they ignore the dependence of each person's contributions on the choices of other group members. For this reason, the regression analysis in Tables 4 and 5 is more pertinent. Here we tested again for sleep treatment effects and various interaction effects. With regressions, we can control for potential differences arising from cohorts 1 and 3, and exploit individual contributions at the round level, rather than averaged over the entire treatment set. Because our main treatment of interest, sleep status, is constant for each subject and group compositions are fixed, we do not used fixed effects but rather pooled cross section regressions and then cluster standard errors to the group level to recognize the interdependence of people's decisions across rounds. To address the lower (zero) and upper (15) censoring of token contributions, we used left and right censored Tobit models, with a comparison OLS specification provided for our sparsest model. We also attempted double hurdle models, but could not get them to converge in even sparse specifications. In all specifications we included round dummies, punishment availability, punishment/non punishment order, and cohort 1 and 3 dummies and their interaction with sleep. To test for sleep treatment effects, we included a dummy for SR status, and an interaction between SR and punishment availability, *PunishAllow*. Our omitted baseline is thus the average contribution of WR subjects in round 1, without punishment. To ask whether sleep affects contributions *without* punishment, we tested the SR coefficient alone. To ask whether sleep affects contributions *with* punishment, we tested the sum of coefficients on SR and SR*PunishAllow*. (With punishment, the total effect for both WR and SR subjects also included the coefficient on *Punishment Allowed*, which cancels out in comparisons.) Though we focus primarily on sleep rather than punishment treatment effects, the coefficient on *PunishAllow* tests how potential punishment affects WR contributions, while the sum of the coefficients on *PunishAllow* and its SR interaction term tests how it affects SR contributions.

In an expanded specification, we add *i*'s lagged (*t-1*) contribution (*Lag Contribution*) as well as the lagged difference of *i*'s contribution from the average of the other two people's contributions (*Lag Deviation of Contribution*). A positive deviation indicates that *i* contributed more than the others. We would expect current contributions to be positively associated with *Lag*

**Table 4. Regression results for contributions—compliant subjects only.**

| *Binary SR WR* | | | | | | | |
|---|---|---|---|---|---|---|---|
| | Treatment Effects | | | | Interaction Effects | | |
| | 1 | 2 | 3 | 4 | 5 | 6 | 7 |
| | OLS | Tobit | Tobit | Tobit | Tobit | Tobit | Tobit |
| Sleep Restricted (SR) | -0.085 | -0.010 | 0.194 | 0.201 | 0.679 | 0.808* | 0.845** |
| | [0.889] | [1.282] | [0.387] | [0.398] | [0.668] | [0.426] | [0.379] |
| SR*PunishAllow | 1.549*** | 2.340*** | 0.728* | 0.770** | | | |
| | [0.562] | [0.865] | [0.382] | [0.372] | | | |
| Punishment Allowed | -0.333 | -0.466 | -0.028 | -0.017 | 0.369* | | |
| | [0.460] | [0.730] | [0.306] | [0.299] | [0.201] | | |
| Lag Contribution | | | 1.274*** | 1.293*** | 1.289*** | 1.322*** | 1.322*** |
| | | | [0.090] | [0.088] | [0.101] | [0.094] | [0.095] |
| Lag Deviation of Contribution | | | -0.488*** | -0.495*** | -0.489*** | -0.537*** | -0.537*** |
| | | | [0.068] | [0.069] | [0.068] | [0.096] | [0.096] |
| Female | | | 0.047 | 0.068 | 0.035 | -0.024 | -0.032 |
| | | | [0.346] | [0.372] | [0.340] | [0.430] | [0.429] |
| Age | | | 0.109*** | 0.111*** | 0.108*** | 0.110*** | 0.109*** |
| | | | [0.033] | [0.036] | [0.033] | [0.034] | [0.035] |
| Cognitive Reflection Test | | | 0.014** | 0.015*** | 0.014** | 0.016** | 0.016** |
| | | | [0.006] | [0.006] | [0.006] | [0.007] | [0.007] |
| SR*Lag Contribution | | | | | -0.019 | | |
| | | | | | [0.089] | | |
| Lag Punishment Received | | | | | | -0.110 | -0.075 |
| | | | | | | [0.112] | [0.202] |
| SR*Lag Punishment Received | | | | | | | -0.053 |
| | | | | | | | [0.249] |
| Constant | 5.600*** | 5.234*** | -5.205*** | -5.335*** | -5.458*** | -4.553*** | -4.553*** |
| | [0.741] | [1.057] | [1.542] | [1.594] | [1.524] | [1.536] | [1.537] |
| | | | | | | | |
| Round dummies | Yes | Yes | Yes | Yes | Yes | Yes | Yes |
| Order dummy | Yes | Yes | Yes | Yes | Yes | Yes | Yes |
| Session 1 dummy | Yes | Yes | Yes | Yes | Yes | Yes | Yes |
| Session 3 dummy | Yes | Yes | Yes | Yes | Yes | Yes | Yes |
| SR*Session 1 | Yes | Yes | Yes | Yes | Yes | Yes | Yes |
| SR*Session 3 | Yes | Yes | Yes | Yes | Yes | Yes | Yes |
| Inverse probability weighting | No | No | No | Yes | No | No | No |
| | | | | | | | |
| N | 2180 | 2180 | 1962 | 1962 | 1962 | 981 | 981 |
| (Pseudo) $R^2$ | 0.059 | 0.011 | 0.143 | 0.148 | 0.143 | 0.164 | 0.164 |
| p value SR+SR*PunAllow = 0: | 0.123 | 0.092 | 0.029 | 0.030 | | | |

Standard errors [in brackets], clustered to group level.

***, **, * denote significance at the 1%, 5%, and 10% levels, respectively, in two tailed tests.

*Contribution* to reflect individual persistence in contributions, and negatively associated with *Lag Deviation of Contribution* if people seek to follow norms or avoid exploitation by others. We also controlled for demographics: gender, age, and score on a 6-item cognitive reflection task. The cognitive reflection task is comprised of 6 questions where heuristic short cuts might

**Table 5. Regression results for contributions—compliant subjects only.**

*Personalized Sleep Deprivation (SD)*

| | Treatment Effects | | | | Interaction Effects | | | |
|---|---|---|---|---|---|---|---|---|
| | 1 | 2 | 3 | 4 | 5 | 6 | 7 | 8 |
| | OLS | Tobit | Tobit | Tobit | Tobit | Tobit | Tobit | Tobit |
| Personalized SD (minutes) | 0.003 | 0.004 | -0.000 | -0.000 | -0.004 | -0.004 | 0.002 | 0.002 |
| | [0.005] | [0.008] | [0.003] | [0.004] | [0.004] | [0.004] | [0.003] | [0.003] |
| SleepDep*PunishAllow | 0.005 | 0.008 | 0.003 | 0.004 | | | | |
| | [0.003] | [0.005] | [0.002] | [0.002] | | | | |
| Punishment Allowed | -0.058 | -0.012 | 0.017 | 0.004 | 0.332 | 0.406** | | |
| | [0.472] | [0.735] | [0.343] | [0.327] | [0.208] | [0.202] | | |
| Lag Contribution | | | 1.287*** | 1.305*** | 1.188*** | 1.189*** | 1.336*** | 1.335*** |
| | | | [0.088] | [0.086] | [0.100] | [0.096] | [0.094] | [0.093] |
| Lag Deviation of Contribution | | | -0. -0.500*** | -0.506*** | -0.498*** | -0.503*** | -0.542*** | -0.540*** |
| | | | [0.068] | [0.069] | [0.069] | [0.069] | [0.097] | [0.096] |
| Female | | | 0.067 | 0.098 | 0.151 | 0.173 | 0.002 | 0.022 |
| | | | [0.357] | [0.387] | [0.368] | [0.390] | [0.444] | [0.436] |
| Age | | | 0.096*** | 0.096*** | 0.093*** | 0.094*** | 0.097*** | 0.102*** |
| | | | [0.032] | [0.035] | [0.031] | [0.033] | [0.033] | [0.037] |
| Cognitive Reflection Test | | | 0.015** | 0.016*** | 0.015** | 0.016*** | 0.017** | 0.018** |
| | | | [0.006] | [0.006] | [0.006] | [0.006] | [0.007] | [0.008] |
| SD*Lag Contribution | | | | | 0.001** | 0.001** | | |
| | | | | | [0.000] | [0.000] | | |
| Lag Punishment Received | | | | | | | -0.049 | 0.535 |
| | | | | | | | [0.103] | [0.728] |
| SD*Lag Punishment Received | | | | | | | | 0.002 |
| | | | | | | | | [0.002] |
| Constant | 5.260*** | 4.824*** | -4.919*** | -4.966*** | -4.489*** | -4.518*** | -4.250*** | -4.324*** |
| | [0.704] | [1.048] | [1.582] | [1.618] | [1.547] | [1.572] | [1.592] | [1.612] |
| Round dummies | Yes | Yes | Yes | Yes | Yes | Yes | Yes | Yes |
| Order dummy | Yes | Yes | Yes | Yes | Yes | Yes | Yes | Yes |
| Session 1 dummy | Yes | Yes | Yes | Yes | Yes | Yes | Yes | Yes |
| Session 3 dummy | Yes | Yes | Yes | Yes | Yes | Yes | Yes | Yes |
| SleepDep*Session 1 | Yes | Yes | Yes | Yes | Yes | Yes | Yes | Yes |
| SleepDep*Session 3 | Yes | Yes | Yes | Yes | Yes | Yes | Yes | Yes |
| Inverse probability weighting | No | No | No | Yes | No | Yes | No | No |
| N | 2180 | 2180 | 1962 | 1962 | 1962 | 1962 | 981 | 981 |
| (Pseudo) $R^2$ | 0.039 | 0.007 | 0.141 | 0.148 | 0.144 | 0.149 | 0.164 | 0.164 |
| p value SD+SD*PunAllow = 0: | 0.107 | 0.121 | 0.313 | 0.291 | | | | |

Standard errors [in brackets], clustered to group level.

***, **, * denote significance at the 1%, 5%, and 10% levels, respectively, respectively, in two tailed tests.

suggest an incorrect answer (see [50]). Our scaled measure created a total score to be in the 0 to 100 range. We opted not to include dummies for group composition of WR and SR subjects, because this composition was unknown to the subjects, and because more direct behavioral measures of its effect that were observable to the subjects are already included, such as the lagged deviation of one's contributions from the group average. Finally, to address attrition of subjects before or during the sleep protocol, we repeat our expanded specification using the

*IPW* selection correction. Results are presented in models 1–4 of Table 4 for compliant subjects, with analogous results for the sample including non-compliant subjects in Table 2 in S1 Appendix.

Results regarding sleep effects are consistent with those of Fig 2 and non-parametric tests. Without punishment, sleep restriction had no significant effect on contributions, as seen by the insignificant coefficient on SR in models 1–4 of Table 4 (or Table 3 in S1 Appendix). With punishment (i.e., *SR+ SR*PunishAllow*), SR had a borderline significant positive effect on contributions. As reported in the final row of Table 4, the two tailed *p*-value on the joint test was *p* = .123 in OLS sparse model 1, *p* = .092 in Tobit sparse model 2, *p* = .029 in full model 3 where lags and demographics are included, and *p* = .030 in the equivalent full model with *IPW* correction. For example, in model (4), SR's contribute on average .971 more tokens than WR's (= 0.201 + 0.770) when punishment was available. Results were similar when including non-compliant subjects (Table 3 in S1 Appendix—in model (4) SR's contributed, on average, .954 more tokens than WR's with punishment; *p* = .018).

To the auxiliary question of whether punishment raised contributions, we found it did for SR's (with joint test of *PunishAllow* plus SR interaction *p*-values of .003, .001, .006 and .002 for models 1–4 of Table 4, or *p* = .005, .002, .007 and .002 for models 1–4 of Table 3 in S1 Appendix), but did not for WR's as evidenced by the insignificance of *PunishAllow* in models 1–4 of either Table.

For robustness, we also tested for sleep effects using *Personal SD*, measured in minutes per night. Results are presented in models 1–4 of Table 5 for compliant subjects, or in Table 4 in S1 Appendix for the full sample. These results confirm that without punishment, the more sleep deprived (relative to a self-assessed optimum) contributed no differently than the less sleep deprived—the coefficient on *Personal SD* was never significant. However, when punishment was available, the evidence that the sleep deprived contributed more generally loses significance, with joint test *p* = .107, .121, .313 and .291 for models 1–4 in Table 5, or *p* = .061, .065, .294, and .266 in Table 4 in S1 Appendix. To illustrate magnitudes in the borderline case of model 1 of Table 4 in S1 Appendix, a 30-min nightly increase in sleep deprivation would be associated with a .27 token (= 30*(.002+.007)) average increase in contributions with punishment available (if *p* = .061 is taken as significant).

To summarize, we found that SR and WR subjects did not differ in public good contributions when peer punishment was unavailable, but some evidence that SR's contributed more when it is. However this second result loses significance when we moved to a quasi-subjective personalized measure of sleep deprivation. Regarding punishment, we found clear evidence it raised contributions for SR's, but not for WR's.

Of equal interest to whether SR's and WR's differed in contribution levels is whether they reacted differently to the (exogenous) introduction of peer punishment, or the (endogenous) behavior of other group members. Graphically, sleep treatment tests asked whether the SR and WR contributions differed in either of the two graphs of Fig 2(A). Differential response tests asked whether SR and WR reactions to the introduction of punishment differed between the two graphs of Fig 2(C). The SR*PunishAllow* interaction term answers the first question in models 1–4 of Table 4 or Table 3 in S1 Appendix. It was significant at the 1%, 5% or 10% levels in all four models of either Table. In particular, the option to punish raised SR contributions by roughly .7 tokens in full models 3 or 4 of Table 4. This effect lost significance when using the continuous *Personal SD* in the analogous four models of Table 5, though it remained significant in the same four models when non-compliant subjects were retained in Table 4 in S1 Appendix.

To test whether SR's and WR's differed in their response to the behavior of other group members, we next introduce other interaction terms in models 5–7 of Table 4 and 5–8 of

Table 5 or Tables 3 and 4 in S1 Appendix. Note that, while we conduct equivalent equal weight and inverse probability weight (*IPW*) corrected specifications for all interaction term regressions, to conserve space we have reported them only when the relevant interaction term was significant in the baseline non-*IPW* corrected model (in practice, the *IPW* correction did not affect the significance of our interaction terms.) First, to test whether SR's are less attentive than WR's to others' evolving contributions, we interacted SR and lagged contribution in model 5 of Table 4 or Table 3 in S1 Appendix, and found it not significant. This remained true in unreported equivalent *IPW* corrected regressions. Again in contrast, however, subjects with greater *Personal SD* had contributions that increased in their own lagged contribution in model 5 of Table 5 or Table 4 in S1 Appendix. This result persisted under the *IPW* correction model 6 of either Table.

Next, to test whether sleep manipulation makes people respond differently to receiving punishment, we introduced *Lagged Punishment Received*, and then an interaction with sleep status in Tables 4, 5, and Tables 3 and 4 in S1 Appendix. (Note that these models can only be run with punishment available, which cuts in half the number of observations). We found in all four tables that subjects did not significantly alter their contributions after being punished in the previous round, and that this lack of response was not affected by sleep status.

To summarize, when testing for differential response, we found that SR's responded to the availability of peer punishment by raising contributions more than do WR's, though the significance of the effect was lost when we moved from binary SR controls to the continuous *Personal SD* measure when using the compliant sample. In contrast, SR's were not more likely to persist in contributing what they had in previous periods, though higher *Personal SD* subjects were. Neither measure of sleep affected the lack of contributions response to being punished.

## Punishment decisions

Because subjects were not aware of the sleep status of the people in their group, we focused on their punishment *sending* decisions without regard to the sleep status of their fellow group members. Fig 3 illustrates the average number of total punishment points sent by compliant subjects to others in their group. When available, SR's appeared to send more punishment

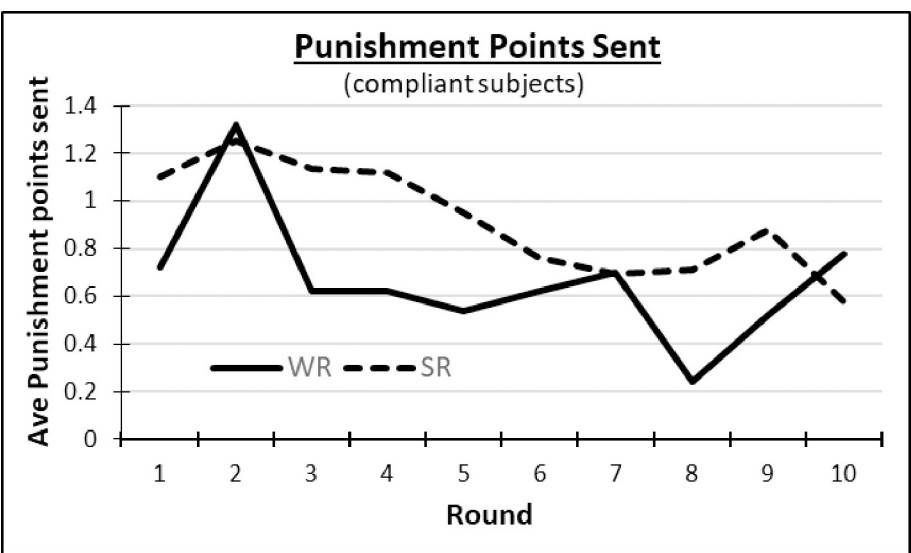

**Fig 3. Average punishment points sent.**

than WR's. To test whether this was significant, we proceed with analysis like that conducted on contributions. Note that subjects were aware of the individual contributions of the other two group members, and could send punishment points to each. While the Veconlab software generated output only of the *total* punishment points sent by each subject, we were able to reconstruct the quantities a subject sent to each individual in almost all cases. In particular, among compliant subjects, we could precisely identify the points a subject sent to each other person for 2136 of 2180 (98.0%) of subject/rounds, or 2464 of 2520 (97.8%) with noncompliant subjects. Where we could not distinguish how punishment points were divided between the other two group members, we imputed an equal division. We retain these imputed cases in the following analysis, but also try excluding them. Results were generally stable to inclusion/exclusion.

Returning to Table 3, the lower panel shows average total punishment points sent by compliant SR's and WR's averaged over 10 rounds, both with or without cohort 1. Analogous results including non-compliant subjects are in Table 1 in S1 Appendix. SR's sent .82 points on average without cohort 1 (or .73 points with it), while WR's sent .54 points (.55 points). Despite appearances, these differences were not statistically significant in 2-tailed Mann Whitney tests on each person's 10-round averaged punishment ($p = .466$ without cohort 1, or $p = .960$ with it).

More formally, we used logit regressions to examine subjects' punishment points sent to another individual, and left (0) and right (10) censored Tobit regressions regarding how many points they sent. We then used interaction terms to test for differential punishment responses of SR's and WR's to the behavior of others. Results using binary SR assignment are presented in Table 6; Table 5 in S1 Appendix, and results using *Personal SD* are in Table 7; Table 6 in S1 Appendix. As with the contributions analysis, we included controls for round, order, cohorts 1 or 3 and their possible interaction with sleep status. Controls were also added to account for a positive deviation of the sender's contribution from the recipient's, and of the absolute value of a negative deviation. We expected subjects to be more likely to punish, and punish more, those with a positive deviation.

Beginning with binary SR controls, again notwithstanding Fig 3, in all models 1–5 in Table 6 (sparse Logit, OLS, and Tobit, full model Tobit and its *IPW*-corrected equivalent) we found no difference in SR's and WR's likelihood of sending punishment, nor the amount sent. Identical results held adding non-compliant subjects in Table 5 in S1 Appendix. Once again, however, results partially changed when using the continuous *Personal SD* measure in Table 7. Here we found in our most credible Tobit models 4–5 that those with higher *Personal SD* sent significantly *fewer* points. In models 4 or 5 of Table 7, for example, an additional 30 minutes of *Personal SD* per night was associated with .27 fewer punishment points being sent (= $30*0.009$). This represents an economically significant 67.5% reduction in punishment sent to other individuals. (The mean punishment sent to another individual by compliant subjects including cohorts 1 and 3 was .40 points.) The effect persisted when non-compliant subjects were added (Table 6 in S1 Appendix). This result is consistent with the hypothesis that subjects who are more sleep-deprived relative to self-assessed optimum were less willing to punish, though we can only speculate whether this was due to reduced altruistic norm enforcement or anger-based punishment.

As an interesting aside, we also found in Tables 6 and 7 that while subjects sent more punishment points to those who contributed *less* than they did, to a lesser extent they also sent more points to those who contributed *more*. The coefficients on the variable |*Neg Dev*| *from Other's Contribution* were smaller than those on the coefficient *Pos Dev from Other's Contribution*, but they were still positive and significant in models 1–5 in both Tables. This non-negligible "anti-social" punishment of higher contributors may explain why peer punishment was

**Table 6. Regression results for punishment sent—compliant subjects only.**

| *Binary SR WR* | | | | | | | | |
|---|---|---|---|---|---|---|---|---|
| | Treatment Effects | | | | Interactions | | | |
| | 1 | 2 | 3 | 4 | 5 | 6 | 7 | 8 |
| | Logit[1] | OLS | Tobit | Tobit | Tobit | Tobit | Tobit | Tobit |
| Sleep Restricted (SR) | 0.002 | 0.162 | 0.331 | 0.254 | 0.173 | -0.656 | -0.827 | 0.047 |
| | [0.047] | [0.115] | [0.835] | [0.746] | [0.743] | [0.913] | [0.898] | [0.717] |
| Pos Deviation Other's Contribution | 0.022*** | 0.133*** | 0.497*** | 0.462*** | 0.475*** | 0.334*** | 0.333*** | 0.461*** |
| | [0.003] | [0.044] | [0.117] | [0.105] | [0.101] | [0.080] | [0.079] | [0.106] |
| \|Neg Deviation\| from Other's Contribution | 0.009** | 0.052** | 0.220** | 0.177** | 0.196** | 0.096 | 0.099 | 0.175** |
| | [0.004] | [0.021] | [0.086] | [0.079] | [0.079] | [0.088] | [0.090] | [0.078] |
| Lagged Punishment Received from Other | | | | 0.890*** | 0.895*** | 0.878*** | 0.882*** | 0.681*** |
| | | | | [0.139] | [0.142] | [0.136] | [0.138] | [0.212] |
| Female | | | | -0.054 | -0.107 | -0.030 | -0.079 | -0.026 |
| | | | | [0.598] | [0.584] | [0.608] | [0.594] | [0.587] |
| Age | | | | 0.021 | 0.028 | 0.026 | 0.033 | 0.023 |
| | | | | [0.077] | [0.077] | [0.077] | [0.078] | [0.078] |
| Cognitive Reflection Test | | | | -0.003 | -0.003 | -0.003 | -0.003 | -0.003 |
| | | | | [0.006] | [0.006] | [0.006] | [0.006] | [0.006] |
| SR*Pos Dev from Other's Contribution | | | | | | 0.208* | 0.219** | |
| | | | | | | [0.110] | [0.106] | |
| SR*\|Neg Dev\| from Other's Contribution | | | | | | 0.178 | 0.199 | |
| | | | | | | [0.130] | [0.128] | |
| SR*Lagged Pun Received from Other | | | | | | | | 0.341 |
| | | | | | | | | [0.270] |
| Constant | Yes | -0.042 | -5.876*** | -7.354*** | -7.695*** | -6.749*** | -7.005*** | -7.305*** |
| | | [0.253] | [1.691] | [2.423] | [2.424] | [2.288] | [2.295] | [2.432] |
| Round dummies | Yes | Yes | Yes | Yes | Yes | Yes | Yes | Yes |
| Order dummy | Yes | Yes | Yes | Yes | Yes | Yes | Yes | Yes |
| Session 1 dummy | Yes | Yes | Yes | Yes | Yes | Yes | Yes | Yes |
| Session 3 dummy | Yes | Yes | Yes | Yes | Yes | Yes | Yes | Yes |
| SR*Session 1 | Yes | Yes | Yes | Yes | Yes | Yes | Yes | Yes |
| SR*Session 3 | Yes | Yes | Yes | Yes | Yes | Yes | Yes | Yes |
| Inverse Probability Weighting | No | No | No | No | Yes | No | Yes | No |
| N | 2180 | 2180 | 2180 | 1962 | 1962 | 1962 | 1962 | 1962 |
| (Pseudo) R$^2$ | 0.133 | 0.166 | 0.094 | 0.131 | 0.141 | 0.134 | 0.145 | 0.132 |

Standard errors [in brackets] clustered to group level.

[1]Average marginal effects dy/dx reported.

***, **, * denote significance at the 1%, 5%, and 10% levels, respectively, respectively, in two tailed tests.

not more effective in raising overall contributions (see Fig 2(B), [14, 15]). We can also identify from the positive and statistically significant coefficient on the *Lagged Punishment Received from Other* variable that subjects generally engaged in counter-punishment of those specific group members who had punished them.

In summary, when focused on sleep treatment effects on punishment sent, we did not find evidence that SR's differed from WR's in their likelihood or quantity of punishing others. Yet we found evidence that the (self-assessed) more sleep-deprived sent less punishment than the less sleep deprived. This effect was robust across Tobit specifications, to the *IPW* selection correction, and to the inclusion of non-compliant subjects in Table 6 in S1 Appendix.

**Table 7. Regression results for punishment sent—compliant subjects only.**

*Personalized Sleep Deprivation (SD)*

| | Treatment Effects | | | | Interactions | | | |
|---|---|---|---|---|---|---|---|---|
| | 1 | 2 | 3 | 4 | 5 | 6 | 7 | 8 |
| | Logit[1] | OLS | Tobit | Tobit | Tobit | Tobit | Tobit | Tobit |
| Personal SD (in minutes) | -0.000* | -0.001 | -0.008** | -0.009** | -0.009** | -0.014*** | -0.014*** | -0.008** |
| | [0.000] | [0.000] | [0.004] | [0.004] | [0.004] | [0.005] | [0.004] | [0.004] |
| Pos Deviation from Other's Contribution | 0.021*** | 0.132*** | 0.483*** | 0.443*** | 0.457*** | 0.380*** | 0.383*** | 0.444*** |
| | [0.003] | [0.045] | [0.119] | [0.106] | [0.101] | [0.144] | [0.149] | [0.106] |
| \|Neg Deviation\| from Other's Contribution | 0.008** | 0.049** | 0.207** | 0.163** | 0.179** | -0.101 | -0.112 | 0.165** |
| | [0.004] | [0.021] | [0.089] | [0.081] | [0.081] | [0.099] | [0.101] | [0.081] |
| Lagged Punishment Received from Other | | | | 0.898*** | 0.897*** | 0.853*** | 0.849*** | 1.001*** |
| | | | | [0.142] | [0.144] | [0.137] | [0.137] | [0.229] |
| Female | | | | -0.062 | -0.064 | -0.114 | -0.109 | -0.056 |
| | | | | [0.606] | [0.581] | [0.611] | [0.582] | [0.604] |
| Age | | | | -0.003 | -0.002 | 0.009 | 0.010 | -0.004 |
| | | | | [0.076] | [0.079] | [0.080] | [0.083] | [0.075] |
| Cognitive Reflection Test | | | | -0.003 | -0.002 | -0.000 | 0.000 | -0.003 |
| | | | | [0.007] | [0.007] | [0.007] | [0.007] | [0.007] |
| SD*Pos Dev from Other's Contribution | | | | | | 0.001 | 0.001 | |
| | | | | | | [0.001] | [0.001] | |
| SD*\|Neg Dev\| from Other's Contribution | | | | | | 0.002*** | 0.002*** | |
| | | | | | | [0.001] | [0.001] | |
| SD*Lagged Pun Received from Other | | | | | | | | -0.001 |
| | | | | | | | | [0.001] |
| Constant | Yes | 0.113 | -4.831*** | -5.949** | -6.196** | -5.636** | -5.804** | -5.943** |
| | | [0.221] | [1.445] | [2.502] | [2.483] | [2.503] | [2.500] | [2.482] |
| Round dummies | Yes | Yes | Yes | Yes | Yes | Yes | Yes | Yes |
| Order dummy | Yes | Yes | Yes | Yes | Yes | Yes | Yes | Yes |
| Session 1 dummy | Yes | Yes | Yes | Yes | Yes | Yes | Yes | Yes |
| Session 3 dummy | Yes | Yes | Yes | Yes | Yes | Yes | Yes | Yes |
| SleepDep*Session 1 | Yes | Yes | Yes | Yes | Yes | Yes | Yes | Yes |
| SleepDep*Session 3 | Yes | Yes | Yes | Yes | Yes | Yes | Yes | Yes |
| Inverse Probability Weighting | No | No | No | No | Yes | No | Yes | No |
| N | 2180 | 2180 | 2180 | 1962 | 1962 | 1962 | 1962 | 1962 |
| (Pseudo) R$^2$ | 0.128 | 0.162 | 0.091 | 0.129 | 0.139 | 0.134 | 0.145 | 0.129 |

Standard errors [in brackets] clustered to group level.

[1]Average marginal effects dy/dx reported.

***, **, * denote significance at the 1%, 5%, and 10% levels, respectively, respectively, in two tailed tests.

Lastly, we used the final three regressions of Tables 6 and 7 to ask whether SR or WR responded differently to the behavior of others when punishing. First, in model 6 of Tables 6 and 7 we included interactions between sleep condition and positive or negative deviation from the other's contribution. Here, we found marginal evidence that SR's sent more punishment than WR's upon learning the other group member contributed *less* to the public good, and evidence that more sleep deprived sent more punishment than less sleep deprived upon learning the other group member contributed *more* to the public good. The positive SR interaction term is significant only at the 10% level in Table 6, while the negative *Personal SD*

interaction term is significant at the 1% level in Table 7. There, in model 6 subjects with an additional 30 minutes of sleep deprivation send 30*.002 = .06 more points to individuals who contributed more than they had. The results were similar with the *IPW*-corrected models (7) in both tables, and they were similar when using the full intent-to-treat samples in Tables 5 and 6 in S1 Appendix.

In sum, even though we found evidence that subjects high in *Personal SD* sent more antisocial punishment, they still sent less punishment overall. In particular, in models 4 or 5 of Table 7, the sum of the main effect on *Personal SD* and the interaction *SD*\*|Neg Dev| was still negative when setting |Neg Dev| at its sample mean (-.014 + (.002*1.78) = -.010).

Finally, to test whether SR's differed from WR's in sending punishment after receiving it in the prior round, we included SR\**Lagged Punishment Received from Other* in model 8 of Table 6. It was not statistically significant. This remained true if using *Personal SD* in Table 7 or if the *IPW*-correction (unreported) was used. Thus, while subjects were more likely to counter-punish those who had punished them, this tendency did not differ by sleep status.

## Conclusion

Surveys and time-use studies have shown that the proportion of adults getting less than 6 hours of sleep per night is increasing. What effects could such mild but persistent sleep restriction have on people's contributions (financial or effort) to public goods for the home, community, or workplace? We examined the effect of a one-week sleep manipulation protocol on subjects' subsequent choices in a voluntary contribution mechanism (VCM) public good experiment, with and without the option of costly peer punishment. We compared the contributions and punishment sending of those who averaged $\leq$ 6.25 hrs/night with those who slept $\geq$ 6.75 hrs/night, thresholds that lie below and above average sleep levels found for adults or participants in other sleep experiments [45]. For robustness, we repeated our analysis using an alternative measure of sleep deprivation relative to personal self-reported optimal sleep, and included those not complying with their assigned sleep protocol, and used inverse probability weighting to address sample attrition during our sleep protocol. Our compliant (N = 109) or full (N = 126) participant sample sizes were less than ideal, but still with sufficient power to detect medium or large size effects ($d \geq$ .556 for N = 109, or d $\geq$ .515 for N = 126) in Mann-Whitney tests, and possibly smaller sized effects in regression analysis.

We did not find that SR subjects were less pro-social than WR subjects regarding contributions, nor more 'trigger happy' in sending costly punishment to others. Rather, absent peer punishment, SR's and WR's did not differ in their contributions. With punishment available, in our most credible full Tobit specifications (with or without the *IPW* correction), SR's contributed significantly more than WR's, and responded more to the introduction of potential punishment. However, these results were not robust to using an alternative partially self-assessed *Personal SD* measure. Similarly, we found no evidence that SR's adjusted their contributions less over time than WR's, but those higher in *Personal SD* did. With greater stability, we found that sleep status did not affect the degree to which people adjusted their contributions in response to receiving punishment.

We next found no evidence that SR's and WR's differed in sending punishment to others, but that those higher in *Personal SD* sent fewer punishment points. Other sleep effects on punishing were also not robust to our choice of sleep measure. We found some evidence (at the 5% or 10% significance levels) that SR's sent more points than WR's to those who had contributed less than they had ("pro-social punishment"), but no equivalent effect when using the *Personal SD* measure. Conversely, we found strong evidence ($p < .01$) that those higher in *Personal SD* sent more punishment to those who had contributed more than they had ("anti-

social punishment"), but no equivalent effect using the binary SR control. Finally, neither sleep measure found any difference in subjects' tendency to "revenge punish".

The fact that a number of our results differ depending on our choice of sleep measure begs the question as to why, and which findings to take as more credible. We examined first the possibility that our sample attrition from the N = 167 registered to N = 126 completed, to N = 109 completed and compliant, somehow plays a role. In particular, our selection regression used to construct the inverse probability weights in Table 2 in S1 Appendix indicated that subjects who reported needing a higher optimal nightly sleep level were significantly more likely to drop out before sleep protocol and VCM task completion. Yet as reported earlier, this drop-out rate was not systematically higher among those initially randomly assigned SR or WR sleep status. Mann-Whitney tests found no significant differences in the distribution of self-reported optimal sleep levels between individuals assigned to SR or WR groups either initially (N = 167, $p$ = .61), nor for those who completed the study (N = 126, $p$ = .53), nor for those who were also found compliant with the sleep protocols (N = 109, $p$ = .61).

An alternative possibility is that the continuous nature of our *Personal SD* measure provided more information than our binary SR/WR assignment, making regression analysis more able to pick up statistically significant differences. Again, we think this an unlikely explanation for our differences because in the six instances in which findings differed between SR/WR analysis and *Personal SD*, three found significance under the SR/WR measure (contributions with punishment, reaction to introduction of punishment, sending more pro-social punishment), and three found significance under *Personal SD* (persistence of contribution level, likelihood/amount of punishment sent, and sending more anti-social punishment). We therefore remain without a satisfying explanation for why results differ to the extent they do by sleep measure. A fuller discussion of the trade-offs between the measures therefore seems warranted.

While both sleep measures are reasonable, we would argue that greater weight should be placed on the results that used the binary SR indicator to control for the subject's sleep state. In other words, we feel more confident in this measure due to our tight control over the random assignment to sleep treatment. And, although a subject may be lost due to attrition over the course of the protocol, we showed both that our results are robust to the use of an *IPW*-correction for selection into the final sample, and that our SR and WR samples did not vary by optimal sleep requirements at all three sample stages (N = 167, 126, or 109). The benefit of using the alternative *Personal SD* control is that it is a continuous measure of one's sleep state and thus may contain additional information. However, there are two noteworthy drawbacks to the *Personal SD* measure. First, it is a composite constructed from both *objective* actigraphy sleep data and *subjective* assessment of "optimal" sleep using whatever internal trade-offs each subject provides. Second, one's degree of compliance represented by the continuous nature of *Personal SD* may be due to other omitted variables over which we did not exercise control (e.g., stress levels during the protocol week).

Accordingly, our firmer results are: 1) SR's contributed no differently than WR's when punishment was unavailable but contributed more when it was, and had a larger contribution reaction to the introduction of peer punishment; 2) SR's did not differ from WR's in their proclivity to punish others generally, though there was marginal evidence that they chose more pro-social punishment of individuals who had contributed less than they had.

These results can be related to previous findings. Dickinson and McElroy [29] and Anderson and Dickinson [27] found *partial* SR reduced subjects' prosocial behaviors, and that *total* sleep deprivation led to choices that limited potential losses in social exchange. While increased VCM contributions may appear cooperative and prosocial, increased contributions only under threat of punishment may instead be aimed at limiting sanction risk. Our results

are thus somewhat consistent with others' findings that SR increases aversion to loss in social exchange.

There are caveats in applying our findings to the field. The social distance between group members in our laboratory VCM was likely greater than that between members of families or work places (see [52]), though comparable to that between donors to charities. Similarly, while a VCM may capture the effects of sleep manipulation on monetary contributions to public goods, further study is needed regarding how real physical effort contributions may be affected by sleep. It is also the case that several design elements of the VCM game we implemented could be varied beyond what we studied in order to pursue other questions. Our design choices were made to fit our study within the extant literature on VCM games with and without punishment so that our main novelty was the sleep manipulation. Nevertheless there are variations in design choice across the vast literature on the VCM game, such as cost of sending or receiving punishment, that may be of interest for future study. Finally, effects could potentially have been more pronounced if the length of sleep manipulation had been prolonged beyond a week, although a full one-week of at-home sleep restriction has been previously shown sufficient to produce the desired effects on sleepiness [37] and certain behavioral outcomes (e.g., [17]). Likewise, we chose to largely remove circadian elements from our design to focus on the impact of sleep restriction, but any compounded effects of suboptimal time of day and sleep restriction could possibly magnify any behavioral effects. As with any experimental design, we faced trade-offs and made choices with regard to our protocol and the decision environment. Some more complex questions may only be answered with additional investigation using variations in protocol and the cooperative dilemma environment.

Overall, we conclude that our results provide no *prima facie* evidence that recent patterns of reduced sleep harm contributions towards group outcomes in a classic cooperative dilemma environment. This is an interesting finding given that these types of environments that pit one's self-interest against group goals (regarding monetary or effort contributions) exist in the home, community, or workplace. The sleep effect we estimated most consistently (across the most specifications in the main text and Appendix sensitivity analysis) showed that, if anything, behavioral "sticks" may motivate the sleepy to cooperate more to avoid the sting of punishment.

## Supporting information

**S1 Appendix. Supplementary appendix Tables and Figure.**
(DOCX)

**S2 Appendix. Experiment instructions (screen shots of Veconlab instructions) NO PUNISHMENT treatment.**
(DOCX)

**S1 File.**
(DO)

**S2 File.**
(DTA)

**S3 File.**
(DTA)

**S4 File.**
(DTA)

## Acknowledgments

We wish to thank two anonymous referees, as well as Andrea Menclova, Abhi Ramalingam, Stephen Knowles, Charles Noussair, Brock Stoddard, and seminar participants at the University of Stirling for valuable comments on the paper.

## Author Contributions

**Conceptualization:** David L. Dickinson.

**Data curation:** Jeremy Clark.

**Formal analysis:** Jeremy Clark.

**Funding acquisition:** David L. Dickinson.

**Investigation:** David L. Dickinson.

**Methodology:** David L. Dickinson.

**Project administration:** David L. Dickinson.

**Resources:** David L. Dickinson.

**Validation:** David L. Dickinson.

**Visualization:** David L. Dickinson.

**Writing – original draft:** Jeremy Clark, David L. Dickinson.

**Writing – review & editing:** Jeremy Clark.

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
