## [Decision Letter · Decision Letter 0]

22 Apr 2020

PONE-D-20-04294

The Effect of Sleep on Public Good Contributions and Punishment: Experimental Evidence.

PLOS ONE

Dear Assoc. Prof. Clark,

Thank you for submitting your manuscript to PLOS ONE. After careful consideration, we feel that it has merit but does not fully meet PLOS ONE’s publication criteria as it currently stands. Therefore, we invite you to submit a revised version of the manuscript that addresses the points raised during the review process.

We would appreciate receiving your revised manuscript by Jun 06 2020 11:59PM. To enhance the reproducibility of your results, we recommend that if applicable you deposit your laboratory protocols in protocols.io, where a protocol can be assigned its own identifier (DOI) such that it can be cited independently in the future. For instructions see: http://journals.plos.org/plosone/s/submission-guidelines#loc-laboratory-protocols

We look forward to receiving your revised manuscript.

Kind regards,

Christian Veauthier, M.D.

Academic Editor

PLOS ONE

Journal Requirements:

Reviewers' comments:

Reviewer's Responses to Questions

**Comments to the Author**

1. Is the manuscript technically sound, and do the data support the conclusions?

Reviewer #1: Yes

2. Has the statistical analysis been performed appropriately and rigorously? 

Reviewer #1: Yes

3. Have the authors made all data underlying the findings in their manuscript fully available?

Reviewer #1: Yes

4. Is the manuscript presented in an intelligible fashion and written in standard English?

Reviewer #1: Yes

5. Review Comments to the Author

Reviewer #1: This paper is about a really important aspect of the potential negative behavioral consequences of sleep deprivation. The study is really well executed and the statistical analysis carefully conducted. The study is written in a quite dense language and is not easily "accessible" for a reader without much prior knowledge of experimental economics. Maybe the manuscript could benefit in the introduction with a less jargon-heavy explanation of the problem this study wants to address.

Overall, there is one major concern regarding the study:

1) Unfortunately the findings really depend strongly on the choice of sleep control variables. The authors suggest more weight should be placed on the SR/WR results because of the objectively measured sleep outcomes. Is this really true? There is a lot of debate about the accuracy of the devices measuring sleep. The authors refer to a 2011 study on the validity of the device. However, is that validity study not outdated? Is there no more recent evidence about the devices' validity? To what extent is the device not just measuring lying in bed rather than actual sleep time. Given the inaccuracy of these devices it would be good to have a more extensive discussion around the limitations associated with this problem.

6. PLOS authors have the option to publish the peer review history of their article (what does this mean?). If published, this will include your full peer review and any attached files.

Reviewer #1: No

---

## [Author Response · Author response to Decision Letter 0]

5 May 2020

April 30th, 2020

Dear Member of the Editorial Board at PLOS ONE and Reviewer of “The Impact of Sleep on Public Good Contributions and Punishment: Experimental Evidence” (PONE-D-20-04294):

First off, co-author David Dickinson and I would like to thank you for the opportunity to re-submit a revision of our manuscript to PLOS ONE for consideration for publication. We were very pleased at this outcome! We would also like to thank the reviewer for his/her considered comments, and for taking the time to carefully read our first submission.

Below I will explain how we have responded to the two concerns raised by the reviewer regarding our manuscript. I quote these concerns verbatim below.

Concern #1: The study is written in a quite dense language and is not easily "accessible" for a reader without much prior knowledge of experimental economics. Maybe the manuscript could benefit in the introduction with a less jargon-heavy explanation of the problem this study wants to address.

Reply #1: Thank you for this comment. We had not adequately adjusted for PLOS ONE being a general science journal, rather than an economics or experimental economics journal. We have re-written various sentences in the Abstract and Introduction so as to start with common language intuition and then transition to labels familiar in economics. To give two examples, rather than starting with “externalities” associated with inadequate sleep, we now talk on p. 1 paragraph 2 of society’s interest in “spillover” effects of inadequate sleep, link this to “external effects”, and finally “externalities”. Similarly, at the opening of the abstract, rather than starting with voluntary contribution mechanism (VCM) experiments, we now speak of “a common public goods experiment,” which we then identify as the voluntary contributions mechanism (VCM). 

We also try to summarize the results of empirical economic research regarding the effects of sleep on labour markets (by Costa-Font and Fléche (2017)) using plainer English.

Concern #2: Unfortunately the findings really depend strongly on the choice of sleep control variables. The authors suggest more weight should be placed on the SR/WR results because of the objectively measured sleep outcomes. Is this really true? There is a lot of debate about the accuracy of the devices measuring sleep. The authors refer to a 2011 study on the validity of the device. However, is that validity study not outdated? Is there no more recent evidence about the devices' validity? To what extent is the device not just measuring lying in bed rather than actual sleep time. Given the inaccuracy of these devices it would be good to have a more extensive discussion around the limitations associated with this problem.

Reply #2: This is a fair question, and one we now try to better address in the paper, and will also summarize here. Just to clarify, the actigraphy devices are not merely “fitbit”-type activity trackers, but rather have been validated across numerous more recent studies in addition to Sadeh (2011). On p. 5 paragraph 1 we have added four more recent studies confirming actigraphy’s ability to distinguish sleep from non-sleep/lack of motion. In addition, we have added in Footnote viii that the use of actigraphy for passive sleep data acquisition is part of the accepted methodology outlined in the joint consensus statement of the American Academy of Sleep Medicine and the Sleep Research Society regarding recommended adult sleep levels (Consensus Conference Panel, 2015).

As we also note in the paper, aside from our SR/WR measure being based on reliable actigraphy sleep measures, it also relies more directly on random assignment than does our other measure.

On the other side of the ledger, two weaknesses of our personalized sleep deprivation measure are that it asks subjects to self-report their optimal sleep (without incentives to think carefully about their answer), and it also requires them to define for themselves what “optimal” means. Some subjects could interpret “optimal” to mean “optimal for physical and mental function, leaving all other considerations aside”, while others could interpret it to mean “optimal for physical and mental function, given the tradeoff they have to make between that function, and having enough awake time to attend to other important considerations each day.” We refer to this argument on par. 2 of p. 24 and footnote xxix.

Thus, while we do believe both measures of sleep status are reasonable enough to include a full discussion of each throughout the paper, in the unfortunate case that they differ in findings, we think there are ex ante good grounds for ultimately treating results from SR/WR assignment as more credible. Yet we hold off to the very end of the paper to do this, enabling a reader who takes a different view to learn of all our results from the other measure.

Thank you again for considering our work.

Sincerely,

Jeremy Clark, 

Associate Professor

Department of Economics and Finance 

University of Canterbury

jeremy.clark@canterbury.ac.nz

---

## [Decision Letter · Decision Letter 1]

16 Jul 2020

PONE-D-20-04294R1

The effect of sleep on public good contributions and punishment: Experimental evidence.

PLOS ONE

Dear Dr. Clark,

Thank you for submitting your manuscript to PLOS ONE. After careful consideration, we feel that it has merit but does not fully meet PLOS ONE’s publication criteria as it currently stands. Therefore, we invite you to submit a revised version of the manuscript that addresses the points raised during the review process. In particular I would like to ask You to discuss more explicitly why some participants were excluded and to respond to all comments made by the reviewers. 

We look forward to receiving your revised manuscript.

Kind regards,

Christian Veauthier, M.D.

Academic Editor

PLOS ONE

Reviewers' comments:

Reviewer's Responses to Questions

**Comments to the Author**

1. If the authors have adequately addressed your comments raised in a previous round of review and you feel that this manuscript is now acceptable for publication, you may indicate that here to bypass the “Comments to the Author” section, enter your conflict of interest statement in the “Confidential to Editor” section, and submit your "Accept" recommendation.

Reviewer #1: (No Response)

Reviewer #2: (No Response)

2. Is the manuscript technically sound, and do the data support the conclusions?

Reviewer #1: Yes

Reviewer #2: Partly

3. Has the statistical analysis been performed appropriately and rigorously? 

Reviewer #1: Yes

Reviewer #2: No

4. Have the authors made all data underlying the findings in their manuscript fully available?

Reviewer #1: Yes

Reviewer #2: Yes

5. Is the manuscript presented in an intelligible fashion and written in standard English?

Reviewer #1: Yes

Reviewer #2: Yes

6. Review Comments to the Author

Reviewer #1: As per initial review, a very interesting study which offers new information on how lack of sleep affects a variety of different aspects in our society.

Reviewer #2: Dear authors,

I deeply apologize for the delay in my report. Attached you find a summary of the paper, my evaluation, and comments.

Let me note that I am not an expert on sleep restrictions or measurement of sleep levels and thus I will mainly comment on the VCM game, the experimental design, methods, and contribution.

As the paper was invited for resubmission before I became a referee, I considered mainly how to improve the current manuscript and recommended to invite you for another revision.

Summary

This paper is concerned with the question of whether a one weeklong mild sleep restriction affects cooperation and punishment behavior in a repeated public good setting (with and without dezentralized punishment). The authors randomly assign participants to a sleep or no sleep restriction condition and compare whether contribution behaviors of people adhering to the sleep restriction differ from those assigned and complying with to the no sleep restriction. The authors identify if at all weak effects of sleep restrictions on cooperation: Using objectively measured sleep outcomes, contributions of sleep restricted individuals are higher, but only if decentralized punishment is available.

Main comments and evaluation

Motivation

The manuscript lacks clear motivation both for why we should consider a mild 1 week sleep restriction treatment instead of a stronger intervention and why we should consider cooperation in a finitely repeated VCM game with and without punishment most relevant. These aspects should be made very clear in the introduction and, concerning the VCM game, also in more detail in the design section (when discussing specific design elements).

Selective attrition

One important aspect of the study and its implications relates to potential selective attrition. The authors needed to exclude several participants in both conditions and discuss this fact only very briefly. However, it could well be that participants dropping out (particularly in the SR condition) bias the observed results. For instance, people complying with the SR condition may suffer less from sleep restrictions than those not complying or dropping out. Hence, differences in behavior may appear minor but would be much larger when such participants are included. I would at least expect a balance table that shows (by treatments) whether dropouts have similar characteristics.

Also I encourage the authors to discuss in the conclusion whether attrition may be a reason for why only minor effects are found. Note: In my view it is perfectly fine if no effect is detected, but the study does not provide a clear null result and appears to have power to identify only “intermediate” or “large”effect sizes. Being clearer about these aspects in the conclusion would make the paper even more transparent and help guiding future research on the topic.

Relatedly, the authors may provide an intention to treat analyses using treatment assignment (ignoring compliance) in the appendix (for those participants who finished the experiment).

Finally, when discussing characteristics of participants, it would be helpful to provide also results from balance tests in Table 1. E.g., there appears to be a higher variance in age in WR, a concern that may be alleviated by showing test results (that may also control for multiple hypotheses testing - but if so, the authors should apply such corrections everywhere).

Design features

The design and different design elements need to be much better motivated. Open questions are:

• Why do the authors study a repeated game?

• Why is punishment designed as a second order public good?

• Why a partner and not a stranger matching?

• Why is punishment implemented as a reduction in points relative to points earned (e.g. 20 percent of the points another group member holds)? This results i) in stronger punishment for “the rich” at the same costs and ii) may add an additional layer of complexity on the game, as it becomes unclear how to coordinate on effective punishment. While the authors implemented a cap on how much points can be reduced, which almost never “bites”, it may still be that punishment decisions are affected by considerations about that cap and whether it might be reached (which may differ across WR/SR groups).

• Why is the id of punisher shown (i.e. why allow for counter punishment)?

• Why did you not inform people about the sleep conditions or assign them to pure WR/SR and mixed groups?

I ask the authors to explain and motivate these design choices more clearly and highlight why these elements are the most relevant when considering the effects of sleep restrictions. The authors may also use some of the questions above to discuss whether the weak effects observed may be a result of some of these choices.

Relatedly, it appears unclear from the manuscript whether participants were aware of both treatment conditions. Please explain in the manuscript whether this was the case and , if it was, why we should not worry about potential “contamination” effects. E.g. higher contributions of SR participants due to a willingness to signal to the experimenter that cooperation is stable even if one was sleep restricted.

Analyses

The way how non-parametric tests are applied appears not meaningful, as it ignores dependencies among group members and individuals. Why not report comparisons based on group level averages? Similarly, the regression analyses do not take into account that individual decisions are not only prone to within subject dependence but also depend on group characteristics (such that error terms may be correlated within groups). Why not use alternative specifications, e.g., individual random effects models with clustering on the group level?

Further, I consider it worthwhile to study counter punishment across treatment conditions as IDs were displayed.

Finally, it may be helpful to add a first round analyses in the appendix showing whether the differences observed occur also initially before learning about others behaviors (i.e. when punishment does not yet include potential counter punishment).

It could also be worthwhile to discuss the robustness of the definition of the non-compliance zone in more detail and/or simply provide ITT estimates in the appendix.

Implications and conclusion:

1) In my reading, the main result is basically: If at all, there are weak effects of sleep restircitions on cooperation behavior in the game considered. As the result is not very clear cut and the study is powered only to identify medium to large effect sizes I advice the authors to avoid making too broad statements such as saying that their results provide no “evidence that recent patterns of reduced sleep are harming voluntary public good funding in the home, community, or workplace” (as they only show that effects are small it in a very specific setting and sample – related to my comments on potential selective attrition).

2) The authors should also discuss more explicitly potential reasons for why the observed effects are small (among them: selective attrition, i.e. those who suffer most do not adhere to SR treatment and thus are excluded, hence they study reactions of people who may cope with sleep restrictions rather well as compared to the average person; exclusion of subjects with strong morning evening types, as they may be affected more / differently; power and identifiable effect size; specifics of the cooperation environment studied, e.g., mixed groups as compared to teams in which all members lack or do not lack sleep; second order public goods effects in punishment condition and complex beliefs about punishment and contribution of others; knowledge of both treatment conditions; etc.)

Minor comments

Page 1 Estimates of the economic costs of insufficient sleep appear very high to me. How reliable are they?

Page 2 “we primarily use binary random assignment (SR vs. WR) to treatment condition,”

The statement can be misleading. The authors study the effects of treatment on the treated and remain silent about potential selective attrition. That is, while treatment assignment is random, the analyses do not focus on intention to treat effects (i.e. comparing those randomly assigned with those not assigned to SR) but measure sleep status and comparing people adhering to the randomly assigned treatment conditions (SR/WR). The authors should thus be more precise here.

Page 3 “They found SR reduced subjects’ prosocial behaviors in general (including trust), which again may have implications for VCM outcomes” Be more precise and mention for which prosocial behaviors effects are found (and for which no effects are observed). This would help readers to put your findings in perspective.

Page 4 “…that sleep deprivation increases people’s aversion to exploitation (or suffering loss in social exchange) might” I think it is quite a stretch to claim that people feel exploited when being punished for not contributing.

Page 6 “allocated between” sounds unclear to me. Participants could decide on how many punishment points to assign to each other group member, right? PLease clarify in manuscript.

Page 8 "However, we could not conduct power analysis for such regression models to identify exact thresholds." You may potentially apply simulation based methods, see for example https://link.springer.com/article/10.1007/s40881-016-0028-4 or https://drive.google.com/file/d/1Neyb97eetVcFD842dUW_kgg_8DrOar6s/view

Page 10: “As foreshadowed, the variable Personal SD has use beyond manipulation checks, as an alternative continuous measure of inadequate sleep (personalized to perceived sleep need). In what follows we shall report results using both approaches. ” The variable “treatment impact” does so as well. It appears unclear why it is ignored here.

Page 10 “While the dominant strategy remained complete free-riding and efficiency full contribution, the exact MPCR differed, as did the complexity of the incentives. For preliminary non-parametric tests, we thus present results with and without cohort 1 (since the exchange rate change does not affect relative incentives), and test whether or not cohort 3 can be pooled with the other cohorts.” The tatement in “()” is only true for selfish decision makers, e.g., inequality averse decision makers tradeoffs could change.

Appendix

Figures in the pdf are not readable (downloaded figures are ok).

7. PLOS authors have the option to publish the peer review history of their article (what does this mean?). If published, this will include your full peer review and any attached files.

Reviewer #1: **Yes: **Marco Hafner

Reviewer #2: No

---

## [Author Response · Author response to Decision Letter 1]

31 Aug 2020

Please see our uploaded Letter to Editor for a summary of our changes in this second revision, and the separate uploaded Reply to Reviewer #2 for a more detailed explanation of our changes.

---

## [Decision Letter · Decision Letter 2]

24 Sep 2020

The effect of sleep on public good contributions and punishment: Experimental evidence.

PONE-D-20-04294R2

Dear Dr. Clark,

We’re pleased to inform you that your manuscript has been judged scientifically suitable for publication and will be formally accepted for publication once it meets all outstanding technical requirements.

Kind regards,

Christian Veauthier, M.D.

Academic Editor

PLOS ONE

Additional Editor Comments (optional):

Reviewers' comments:

Reviewer's Responses to Questions

**Comments to the Author**

1. If the authors have adequately addressed your comments raised in a previous round of review and you feel that this manuscript is now acceptable for publication, you may indicate that here to bypass the “Comments to the Author” section, enter your conflict of interest statement in the “Confidential to Editor” section, and submit your "Accept" recommendation.

Reviewer #2: All comments have been addressed

2. Is the manuscript technically sound, and do the data support the conclusions?

Reviewer #2: (No Response)

3. Has the statistical analysis been performed appropriately and rigorously? 

Reviewer #2: (No Response)

4. Have the authors made all data underlying the findings in their manuscript fully available?

Reviewer #2: (No Response)

5. Is the manuscript presented in an intelligible fashion and written in standard English?

Reviewer #2: (No Response)

6. Review Comments to the Author

Reviewer #2: (No Response)

7. PLOS authors have the option to publish the peer review history of their article (what does this mean?). If published, this will include your full peer review and any attached files.

Reviewer #2: No

---

## [Editor Report · Acceptance letter]

8 Oct 2020

PONE-D-20-04294R2 

The effect of sleep on public good contributions and punishment:  Experimental evidence. 

Dear Dr. Clark:

I'm pleased to inform you that your manuscript has been deemed suitable for publication in PLOS ONE. Congratulations! Your manuscript is now with our production department. 

Kind regards, 

on behalf of

Dr. Christian Veauthier 

Academic Editor

PLOS ONE